# Identification and functional modelling of plausibly causative *cis*-regulatory variants in a highly-selected cohort with X-linked intellectual disability

Hemant Bengani[1‡], Detelina Grozeva[2,3‡], Lambert Moyon[4‡], Shipra Bhatia[1‡], Susana R. Louros[5,6], Jilly Hope[7], Adam Jackson[5], James G. Prendergast[8], Liusaidh J. Owen[1], Magali Naville[4], Jacqueline Rainger[1], Graeme Grimes[7], Mihail Halachev[7], Laura C. Murphy[7], Olivera Spasic-Boskovic[9], Veronica van Heyningen[1], Peter Kind[5,6], Catherine M. Abbott[6,7], Emily Osterweil[5,6], F. Lucy Raymond[2‡], Hugues Roest Crollius[4‡], David R. FitzPatrick[1,6‡]*

1 MRC Human Genetics Unit, IGMM, University of Edinburgh (UoE), Edinburgh, United Kingdom, 2 Cambridge Institute for Medical Research, University of Cambridge, Cambridge, United Kingdom, 3 Institute of Psychological Medicine & Clinical Neurosciences, Cardiff University, Cardiff, United Kingdom, 4 Ecole Normale Supérieure, Institut de Biologie de l'ENS, IBENS, Paris, France, 5 Centre for Discovery Brain Sciences, Patrick Wild Centre, University of Edinburgh, Edinburgh, United Kingdom, 6 Simons Initiative for the Developing Brain, University of Edinburgh, Edinburgh, United Kingdom, 7 Institute of Genomic and Molecular Medicine, University of Edinburgh, Edinburgh, United Kingdom, 8 Roslin Institute, University of Edinburgh, Edinburgh, United Kingdom, 9 East Midlands and East of England NHS Genomic Laboratory Hub, Molecular Genetics, Adden brooke's Hospital, Cambridge University Hospitals NHS Foundation Trust Cambridge, Cambridge, United Kingdom

‡ HB, DG, LM and SB share first authorship on this work. FLR, HRC and DRF are joint senior authors on this work
* david.fitzpatrick@hgu.mrc.ac.uk, David.FitzPatrick@igmm.ed.ac.uk

**Data Availability Statement:** All relevant data are within the paper and its Supporting Information files. RNAseq data files can be found in the NCBI

## Abstract

Identifying causative variants in *cis*-regulatory elements (CRE) in neurodevelopmental disorders has proven challenging. We have used *in vivo* functional analyses to categorize rigorously filtered CRE variants in a clinical cohort that is plausibly enriched for causative CRE mutations: 48 unrelated males with a family history consistent with X-linked intellectual disability (XLID) in whom no detectable cause could be identified in the coding regions of the X chromosome (chrX). Targeted sequencing of all chrX CRE identified six rare variants in five affected individuals that altered conserved bases in CRE targeting known XLID genes and segregated appropriately in families. Two of these variants, *FMR1*CRE and *TENM1*CRE, showed consistent site- and stage-specific differences of enhancer function in the developing zebrafish brain using dual-color fluorescent reporter assay. Mouse models were created for both variants. In male mice *Fmr1*CRE induced alterations in neurodevelopmental *Fmr1* expression, olfactory behavior and neurophysiological indicators of FMRP function. The absence of another likely causative variant on whole genome sequencing further supported *FMR1*CRE as the likely basis of the XLID in this family. *Tenm1*CRE mice showed no phenotypic anomalies. Following the release of gnomAD 2.1, reanalysis showed that *TENM1*CRE exceeded the maximum plausible population frequency of a XLID causative allele. Assigning causative status to any ultra-rare CRE variant remains problematic and requires

Gene Expression Omnibus (accession number GSE180066).

**Funding:** DRF and VvH were supported by MRC University Unit grant to the MRC Human Genetics Unit at the University of Edinburgh. HB & MN and project costs were supported and funded by the 7th framework programme of the European Union [NeuroXsys Project HEALTH- F4-2009-223262]. HB was subsequently funded by a grant from NewLife (Grant Ref: 14-15/07). National Institute of Health Research Bioresource for Rare Diseases (grant number RG65966) for whole genome sequence data from 12,596 X chromosome alleles as controls. JH is funded by a BBSRC studentship. FLR and DG are funded by NIHR Cambridge Biomedical Research Centre grant. HRC received support from the French Government from programs implemented by ANR with the references ANR–10–LABX–54 MEMOLIFE and ANR–10–IDEX–0001–02 PSL∗ Research University. PK received support from Simons Initiative for the Developing Brain Simons Foundation (US). The funders had no role in study design, data collection and analysis, decision to publish, or preparation of the manuscript.

**Competing interests:** No authors have competing interests

disease-relevant in vivo functional data from multiple sources. The sequential and bespoke nature of such analyses renders them time-consuming and challenging to scale for routine clinical use.

## Introduction

*Cis*-regulatory elements (CRE; encompassing enhancers and repressors) are genomic sequences that control transcriptional activity of one or more genes on the same chromosome *via* sequence-specific interaction of the DNA with proteins and/or RNA. CRE can be predicted using comparative genomics [1], transcriptional characteristics [2], patterns of histone modifications and protein association [3], patterns of accessible chromatin [4] and direct interactions with promoters [5]. Although estimates of the number of CRE in the human genome vary with each prediction method, functional ENCODE data has been interpreted as identifying at least 400,000 putative human enhancers [6]. Disrupted CRE function as a cause of Mendelian disease was first recognized *via* the loss or gain of regulatory function resulting from structural chromosome anomalies such as deletion or translocation [7–10]. However, the identification of disease-associated single nucleotide variants within individual CRE has been complicated by several factors. CRE can function over large genomic intervals and the targeted gene may not be the closest gene. CRE mostly exist in the non-coding parts of the human genome where our current understanding of mutation consequence is very incomplete compared to the coding region.

Developmental disorders (DD) are a diverse group of conditions caused by perturbations of embryogenesis or early brain development. The combination of massively parallel sequencing technologies and family-based analyses has proven very effective in identifying the genes and mechanisms causing severe developmental disorders in humans. DD are primarily genetically determined with a high proportion of causative coding region variants arising as *de novo* mutations (DNM) [11]. The genomic intervals encompassing known DD causative genes are commonly enriched in highly conserved CRE [12]. DNM enrichment is also evident in evolutionarily conserved, brain-active CRE in severe DD at a cohort level [13] but the confident assignment of variants as causative in affected individuals is not yet possible [14].

We have previously identified all likely CRE on the human X chromosome and assigned these to their target genes [15]. Here we have sequenced all of these CRE in 48 individuals with intellectual disability (ID) and a family history indicating that the ID is X-linked (XLID). Each affected individual had previously had a negative screen for likely causative mutations in all coding exons on the X chromosome [16]. Following strict filtering, six rare variants in CRE predicted to control known XLID genes were tested *in vivo* using zebrafish and mouse models to classify their diagnostic potential. After these studies and reanalysis of the population allele frequencies following the release of gnomAD 2.1 data, only one CRE variant, causing a complex dysregulation of the gene *FMR1*(FMRP translational regulator 1), could be considered as likely causative in a single family.

## Materials and methods

### Cohort selection

Genomic DNA samples from 48 individuals (probands) with moderate-to-severe intellectual disability (ID) were used in this study. Research ethics review and approval was granted by the

UK Multicentre Research Ethics Committee in Cambridge with approval number 03/0/014. Written consent was obtained from the parents or guardians of each affected individual included in the study. Each individual is assumed to have X-linked recessive form of ID on the basis of positive family history: three or more cases of ID in males only, predominant sparing of carrier females and no evidence of male-to-male transmission of the disease. A clinical geneticist had assessed the individuals and the cause of the ID was unknown. The severity of the disease was categorized using DSM–IV or ICD-10 classifications (profound mental retardation was classified as severe). The affected individuals had previously been tested negative by routine diagnostic approaches (i.e., CGH microarray analysis at 500 kb resolution, fragile X [MIM 300624], methylation status of Prader Willi [MIM 176270]/Angelman syndrome [MIM 105830]). In addition, all 48 individuals have been screened within a previous study [16] for coding variants on the X chromosome likely to lead to disease and such variants had not been found. The whole genome sequencing of individual S3 was performed and analyzed within the UK National Health Service as part of a large-scale clinical implementation study led by one of the authors (FLR) [17].

## Targeted capture design and sequencing

A comprehensive list of coordinates of all exonic and conserved regulatory elements from chrX used to design a customized capture library (Roche, NimbleGen) is provided in **S1 Table**. Library preparation, pre- and post-capture multiplexing were performed using the Seq-Cap EZ Choice XL kit (Roche NimbleGen) and TruSeq index barcodes (Illumina) were used according to the manufacturer's instructions. 4 different DNA samples were pooled for pre-capture multiplexing and 4 post-captured libraries were combined. Paired-end sequenced performed on a single lane of a HiSeq-2000 instrument (Illumina). In total 16 different DNA samples were sequenced in a single lane of a HiSeq-2000 and 4 lanes were used to sequence all 48 DNA samples.

## Read mapping, variant analysis and enhancer selection

Following quality control with FastQC, reads were mapped to the GRCh37 version of the human reference genome using BWA [18]. Variants were called using GATK [19] according to its recommended best practice pipeline. 40,699 variants remained after filtering out variants that failed GATK's variant quality score recalibration. These variants were subsequently compared to dbSNP v137 to filter out common variants. Any variant with one of the following handles in dbSNP (1000GENOMES, CSHL-HAPMAP, EGP_SNPS, NHLBI-ESP, PGA-UW-FHCRC) were excluded where the variant's reported minor allele frequency was greater than 0.01 or the minor allele was observed in at least two samples. The remaining 9,577 chrX variants were then annotated with SnpEff [20] to determine their predicted effects on genes.

To determine the best candidates for experimental validations, the variants were ranked based on extreme evolutionary conservation. Using Multiple Sequence Alignments from 45 vertebrate species against the Human genome (UCSC genome browser), mutations were retained if the reference human allele was conserved in at least 90% of the species, and then sorted by decreasing conservation depth. Top variants were then manually evaluated using biochemical signals from the ENCODE project (H3K4me1, H3K4me3, H3K27ac, DNase1 sensitivity), and based on the association to target genes known to be responsible for XLID or functionally related to brain development, leading to a final selection of 31 candidate variants (**S2 Table in S1 File**). Target genes for each of the CRE harboring the variants were assigned as described previously [15].

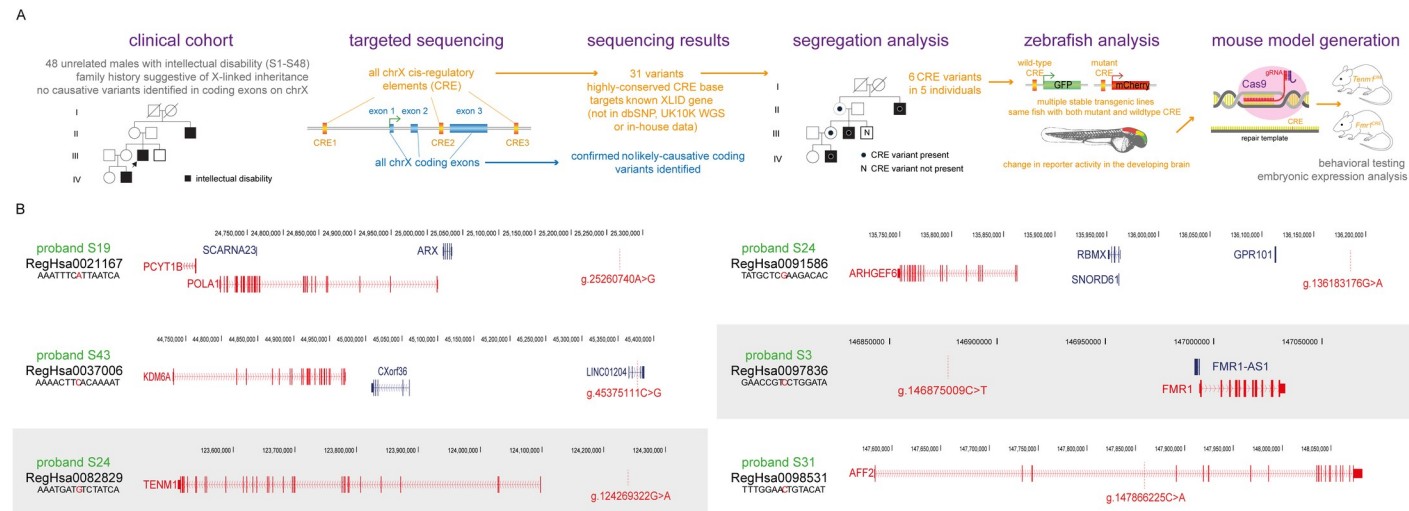

**Fig 1. Project summary and XLID-associated regulatory variants and their predicted target genes.** (A) A diagrammatic summary of the experimental pipeline followed in this paper. (B) Schematic showing the genomic region of the six genomic variants in the five probands (S19, S24, S43, S3 and S31) indicating the location of the XLID-associated CRE variants along with their predicted target genes indicated in red, genomic coordinates from h19/GRCh37 genome build. The variants highlighted by grey box were used to make mouse models.

Motif search on CRE element was performed on a 40bp window around the mutated base for both human and mouse sequences using the FIMO software from the MEME suite [21]. The motif databases used for the search were Jaspar Core 2018 for vertebrates and Uniprobe mouse motifs as downloaded from the MEME website. Motifs with a p-value of 0.001 or lower that were present uniquely in either the WT or the mutant sequences are reported.

## Animal study licenses

All mouse and zebrafish experiments were approved by The University of Edinburgh ethical committee and performed under UK Home Office license number PIL 60/12763, 70/25905, I655D57B6, PA3527EC3 and 1724D1B2C; PPL 60/4418, 60/4424, IFC719EAD and 60/4290.

## Transgenic zebrafish, In Situ Hybridization (ISH) and morphant generation

The wild-type and mutant versions of the six variants documented in **Fig 1** and **Table 1** were analyzed for their regulatory activities in dual color enhancer-reporter transgenic assays in zebrafish embryos [22]. The sequences of the primers used in generating the constructs

**Table 1. Allele frequencies of the six variants tested in the dual-color reporter transgenic assays.**

| Proband | Variant | Target Gene | Alleles | Total Alleles | Hemizygotes | Allele Frequency* |
|---------|---------|-------------|---------|---------------|-------------|-------------------|
| S3 | X-146875009-C-T | *FMR1* | 0 | NA | 0 | NA |
| S19 | X-25260740-A-G | *POLA1 PCYT1B* | 0 | 21967 | 0 | NA |
| S24 | X-124269322-G-A | *TENM1* | 1 | 21962 | 0 | 0.00004553 |
| S24 | X-136183176-G-A | *ARHGEF6* | 7 | 21979 | 1 | 0.0003185 |
| S31 | X-147866225-C-A | *AFF2* | 9 | 22020 | 3 | 0.0004319 |
| S43 | X-45375111-C-G | *KDM6A* | 0 | NA | 0 | NA |

*Popmax Filtering AF (95% confidence).

utilized in the assay are listed in **S3 Table in S1 File.** The number of independent lines analyzed for each enhancer and their expression sites is summarized in **Table 2.** The transgenic F1 embryos were processed for imaging as described [22]. The images were taken on a Nikon A1R confocal microscope and processed using A1R analysis software.

A zebrafish *six3* antisense morpholino oligonucleotide (Six3AMO) was obtained from Gene Tools, LLC, with the following sequence: 5´ GCTCTAAAGGAGACCTGAAAACCAT 3´. This morpholino has sequence complementary to the highly conserved sequences around the translation initiation codon of both *six3a* and *six3b*, and hence inhibits the function of both zebrafish *six3* genes [23]. As control we used the Gene Tools LLC standard negative control morpholino: 5´ CCTCTTACCTCAGTTACAATTTATA 3´. The morpholinos were injected into 1 to 2-cell stage of at least 100 embryos to deliver an approximate amount of 2.5 ng per embryo.

RNA in situ hybridization on fish embryos was performed as previously [24]. The sequences of primers used for synthesis of specific probes are listed in (**S3 Table in S1 File).**

## Generation of transgenic mice and embryo ISH

CRISPR/Cas9 gene targeting technology was used to generate mouse lines with orthologous mutations; *Fmr1*CRE and *Tenm1*CRE (Teneurin transmembrane protein 1). A double-stranded DNA oligomer that provides a template for the guide RNA sequence was cloned into px461. The details of guide RNA and repair template sequence are provided in **S1 Note in S1 File.** The full gRNA template sequence was amplified from the resulting px461 clone using universal reverse primer and T7 tagged forward primers. The guide RNA was generated from this PCR template using T7 RNA polymerase (NEB), and purified with RNeasy mini kit (Qiagen) purification columns. The zygotic injection mix contained Cas9 mRNA (Tebu Bioscience @ 50ng/µl), guide RNA (25ng/µl) and repair template single stranded DNA (IDT 150ng/µl). Injected embryos were transferred into the oviducts of pseudo-pregnant females to litter down. Genotyping of the resulting mice was performed by Sanger sequencing using tail tip DNAs. F0 mice with desired variant were crossed with C57BL/6 to generate a stable mice line.

*In situ* hybridization on mouse embryos was performed with DIG-labelled gene-specific antisense probes as previously described [25]. The sequences of primers used for synthesis of specific probes are listed in **S3 Table in S1 File.**

## Olfaction test

Male wild-type and *Fmr1*CRE and *Tenm1*CRE littermates at P25 were subjected to the buried food test assay. For three consecutive days before testing ¼ chocolate button (Cadbury) was placed in the home cage for 15 minutes to habituate the mice to the food reward. 12 hours before the test, all food was removed from the home cage to motivate the mouse to find the food reward during the test. After 12 hours, the mouse was placed in a clean cage with fresh bedding in which ¼ chocolate button had been buried 1cm beneath the bedding. The time taken to find the buried food was scored and the test was stopped if the mouse did not find the food after 15 minutes. The bedding was replaced and the cage cleaned with 1% Conficlean between mice. All mice were scored blind to the genotype. Unpaired t-tests were used to determine statistical significance.

## Seizure propensity testing of *Fmr1*CRE

Male wild-type and *Fmr1*CRE littermates at P25 were tested for audiogenic seizures as described previously [26]. Briefly, animals were transferred to a transparent plastic test chamber and, after 1 minute of habituation, exposed to a 2 min sampling of a modified personal alarm held

**Table 2. Analysis of stable transgenic lines of dual-color reporters in zebrafish embryos.**

| Target Gene | No. stable lines analysed | Mutation Status | Reporter | Olfactory placode | Forebrain | Trigeminal ganglia | Midbrain | Hindbrain | Neural Tube | Lateral spinal cord neurons | Eye | Otic vescle | Heart | Pectoral fin | Effect of CRE Variant |
|---|---|---|---|---|---|---|---|---|---|---|---|---|---|---|---|
| *TENM1* | 4 | Wild-type | eGFP | | | | 4 | 4 | 4 | | | | 1 | 2 | Loss of midbrain and hindbrain activity |
| | | Mutant | mCherry | 1 | | | 0 | 0 | 4 | | | 1 | | | |
| *FMR1* | 3 | Wild-type | eGFP | | 3 | 3 | | | | 3 | | | 1 | | Loss of forebrain activity |
| | | Mutant | mCherry | | 0 | 3 | | | 1 | 3 | | | | | |
| *POLA1-PCYT1B* | 3 | Wild-type | eGFP | | 1 | | | 1 | 1 | | 1 | | | | Uninterpretable |
| | | Mutant | mCherry | | 1 | | | 1 | 1 | | | 1 | | | |
| *ARHGEF6* | 3 | Wild-type | eGFP | | 1 | | 1 | 1 | 1 | | | | | | Uninterpretable |
| | | Mutant | mCherry | | 1 | | 1 | 1 | 2 | | | | | 1 | |
| *KDM6A* | 3 | Wild-type | mCherry | | | | 1 | 1 | 2 | | | | | 1 | Uninterpretable |
| | | Mutant | eGFP | | | | 1 | 1 | 2 | | 1 | | | 2 | |
| *AFF2* | 2 | Wild-type | mCherry | | 1 | | 1 | 1 | | | | | | | Uninterpretable |
| | | Mutant | eGFP | | 1 | | 1 | 1 | | | | | | | |

at > 130dB. Seizures were scored for incidence (seizure/no seizure) and severity, with an increasing scale of 1 = wild running, 2 = clonic seizure, and 3 = tonic seizure. All mice were tested and scored blind to genotype. Statistical significance for incidence was determined using two-tailed Fisher's exact test.

### Basal protein synthesis and FMRP western blotting

Protein synthesis levels were measured following the protocol outlined by Osterweil [27]. The detailed protocol is described in **S2 Note in S1 File**.

For western blots, hippocampal slice from P25 male wild-type and *Fmr1*^CRE knock-in mutant littermates were dissected and homogenized in lysis buffer (20 mM HEPES pH 7.4, 0.5% Triton X-100, 150 mM NaCl, 10% glycerol, 5 mM EDTA with protease inhibitor cocktail (Roche), incubated at 4°C for 30 min followed by centrifugation at 14000 rpm for 30 min to collect the supernatant. These samples were directly used for SDS-PAGE and transferred onto nitrocellulose membranes for immunoblot analysis with FMR1 antibody (MAB2160, Milipore). Densitometry was performed on scanned blot film using Image Studio Lite software. Each signal was normalized to total protein in the same blot. Values are shown as a percentage of average WT for graphical purposes.

### Hippocampal slice electrophysiology

Electrophysiology experiments were performed as described [28]. The detailed protocol is described in **S3 Note in S1 File**.

### RNAseq and RNAscope analysis

*In situ* RNA hybridization was performed using the RNAscope assay (Advanced Cell Diagnostics, ACD, Hayward, CA, USA) according to the manufacturer's recommendations. The detailed protocols described in the **S4 Note in S1 File**. The images of sections were processed using the multimodal Imaging Platform Dragonfly (Andor Technologies, Belfast, UK) using air 40x Plan Fluor 0.75 DIC N2. Data were collected in Spinning Disk 25 μm pinhole mode on the high sensitivity iXon888 EMCCD camera. According to Advanced Cell Diagnostics, each mRNA molecule hybridized to a probe appears as separate small puncta. Data visualization and spot counting was done using IMARIS 8.4 (Bitplane). The details of the RNAseq analysis are given in **S5 Note in S1 File.**

### Statistical analysis

Statistical analysis was performed using two-tailed Student's *t*-test (Prism 4, GraphPad Software, La Jolla, CA, USA) except for Fig 6C were significance was determined using two-tailed Fisher's exact test (appropriate for analyzing nominal data sets). A p-value of < 0.05 was considered statistically significant. Data are shown as the mean values ± SE of number of replicates (n) used in the experiments.

## Results

### Identifying a cohort likely to be enriched in disease-associated CRE

In a previous study we found that 155/208 families with apparent XLID had no detectable disease-associated variants in the coding sequence on the X chromosome [16]. We chose affected male probands from 48 of these undiagnosed families for inclusion in present study. Each unrelated proband had 3 or more similarly affected relatives with the inheritance pattern being strongly suggestive of XLID. We reasoned that these families are likely to be enriched for

highly penetrant causative regulatory mutations. In addition, the high prior probability that any causative variant in these families would be located on the X chromosome significantly reduced the genomic space for interrogation. A summary of the overall study design is presented in **Fig 1A**.

## Identifying rare variants in CRE on the X chromosome

We performed targeted sequencing in each proband using a custom 15.9 Mb oligonucleotide pull-down consisting of 227,323 baits. These baits were designed to capture two non-overlapping sets of target sequences from the human X chromosome (chrX); all chrX coding exons and all chrX CRE. The set of chrX CRE, accounting for 4.4% of chrX genomic sequence, had been defined in a previous study [15]. This study also showed that the maintenance of linkage between a CRE and neighbouring genes throughout evolution was an accurate way to identify the target gene(s). Target genes assigned using this conserved synteny approach allowed a maximum CRE to gene distance of 1.5 Mb [15]. Approximately a third of chrX CRE could be assigned to a single gene with the remainder having >1 equally plausible target. 389/812 protein coding genes on the X chromosome could be assigned to at least one CRE.

Following sequencing and alignment, a total of 40,699 variant calls passed basic quality controls in these individuals (**S1 Fig in S1 File**). As expected from our previous work [16], no likely causative variants were identified in the coding exons. 628 hemizygous variants were identified in high confidence putative CRE and were not present in the population-based whole genome sequence data that was available at the time (**S1 Fig in S1 File**). To further increase the likelihood of identifying clinically-interpretable variants we focused on the 31/628 altered highly conserved bases in CRE that were predicted to control known XLID genes. 30/31 were confirmed by Sanger sequence analysis in the probands. 6 of these variants were shown to segregate appropriately in the XLID families using samples from additional affected, unaffected males and obligate female carriers (**Fig 1B**). Details of segregation in available family members are shown in **S2 Table** and **S3-S33 Figs in S1 File**. 4/48 probands carried one of these six variants and 1/48 carried two.

## *FMR1*$^{CRE}$ and *TENM1*$^{CRE}$ CRE variants alter enhancer function in zebrafish transgenics

The reference and alternative base versions of all six CRE variants were then tested for CRE function using a dual-color fluorescent transgenic assay in zebrafish [22]. Multiple stable transgenic lines were created in which the wild-type and mutant human CRE drives expression of different fluorescent proteins in the same fish (**Figs 2A** and **3A**). Reporter expression domains were scored in living embryos between 24 hours and 96 hours post-fertilization (hpf). Only consistent differences between the reference and alternative alleles in at least 3 independent lines were taken as evidence of a functional effect of the mutation. The specific criteria for a variant to be included in future *in vivo* functional studies were: 1. Strong evidence of variant-associated disruption of CRE activity in the developing brain. 2. A significant overlap between the wild-type CRE activity and that of the endogenous neural expression of the orthologous zebrafish gene (**Figs 2C** and **3C**).

Only two CRE variants in two different probands fulfilled these criteria (**Table 2, S34-S39 Figs in S1 File**): *TENM1*$^{CRE}$ (proband S24) and *FMR1*$^{CRE}$ (proband S3). *TENM1*$^{CRE}$ showed a loss of reported expression in the mid- and hind-brain (**Fig 2D and 2E**). *FMR1*$^{CRE}$ resulted in the loss of expression in the forebrain but normal expression in the trigeminal ganglia (**Fig 3D and 3E**).

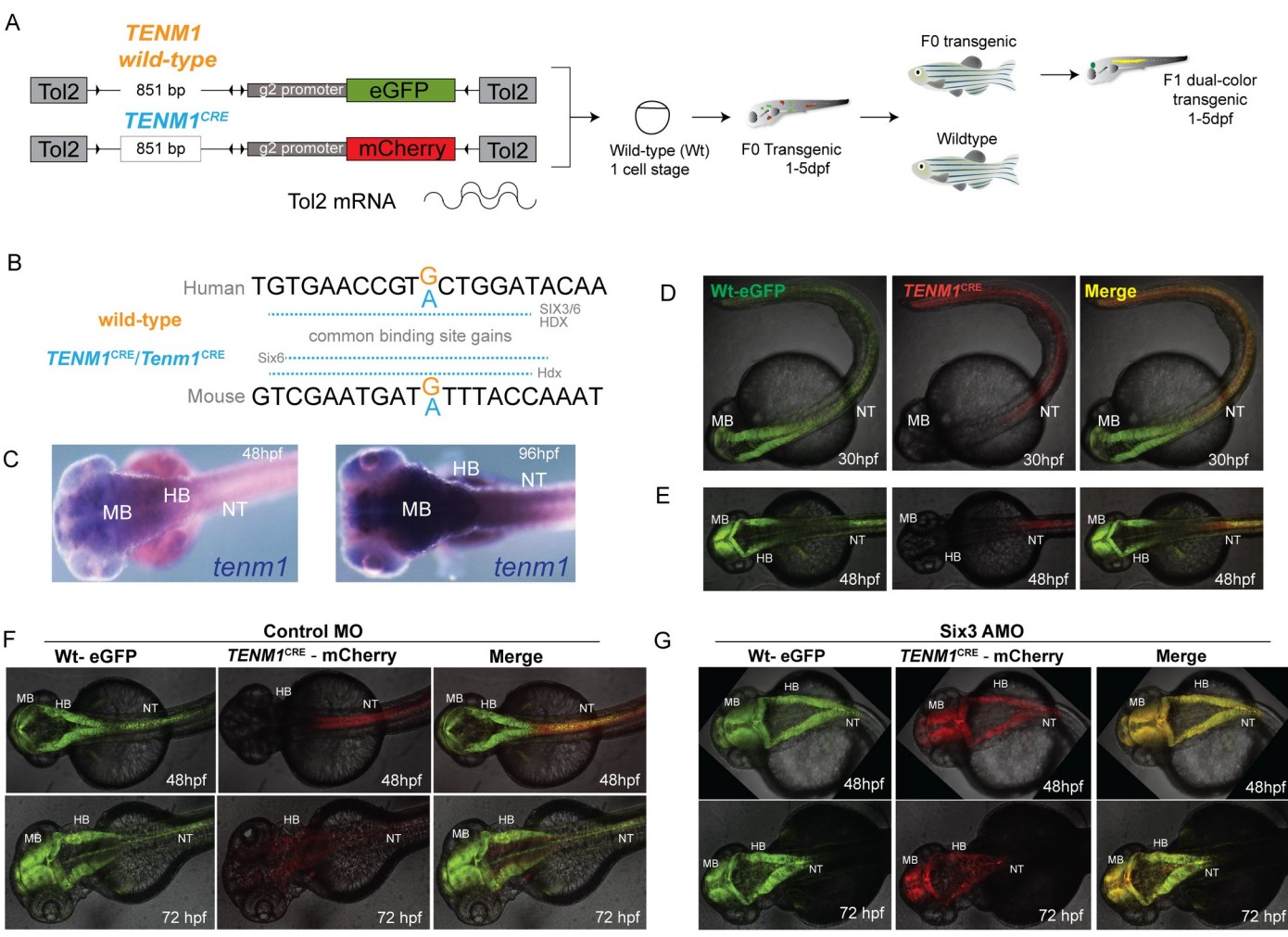

**Fig 2. *TENM1*CRE alters enhancer function in the zebrafish brain by creating a repressive SIX3 binding site.** (A) A diagrammatic summary of the dual color fluorescence assay used in this study. The size of the human *TENM1* element is provided in the left hand panel in base pairs (bp) (B) Human and mouse (*TENM1*CRE/*Tenm1*CRE) sequences are shown with the variant base marked in blue, resulting in gain of SIX3/SIX6 and HDX binding sites in *TENM1*CRE and Six6 and Hdx binding sites in *Tenm1*CRE. (C) *mRNA in situ* hybridization showing expression of *tenm1* in midbrain, hindbrain and neural tube during embryonic development in wild-type zebrafish. (D-E) Dual color fluorescent transgenic assay in zebrafish with wild-type (Wt) and mutant *TENM1*CRE driving eGFP and mCherry expression respectively. Loss of enhancer activity is observed in midbrain and hindbrain with the mutant *TENM1*CRE allele. Further examples of embryos for different stable lines are shown in **S34 Fig in S1 File**. (F-E) *six3* knockdown rescues the effect of the mutant variant on the activity of *TENM1*CRE. Control morpholino injected embryos show loss of reporter activity in midbrain and hindbrain by mutant allele, where the mutation creates a Six3 binding site (E). Knockdown of *Six3* rescues the activity of mutant allele in the midbrain and hindbrain (F). MB: Midbrain; HB: Hindbrain; NT: Neural tube; hpf: Hours post fertilization.

## Transcription factor binding site analysis of CRE variants

We next looked at the effect of these CRE variants on putative transcription factor binding sites. We restricted the analysis to sites that were gained or lost in both the human and mouse versions of the CRE. The *TENM1*CRE variant created a novel site predicted to bind SIX3 (SIX homeobox 3) or SIX6 (SIX homeobox 6) in both the human and orthologous mouse CRE (**Fig 2B**). SIX3 is essential for early brain development and has pathway-specific activator and repressor activity [29]. To determine if SIX3-mediated repression may be responsible for the altered enhancer activity in the variant *TENM1*CRE we chose to use morpholino-induced knock-down of endogenous *six3* in *TENM1*WT/*TENM1*CRE transgenic embryos. The phenotypic effect of the morpholinos targeting zebrafish *six3* was assessed by 1-cell embryo

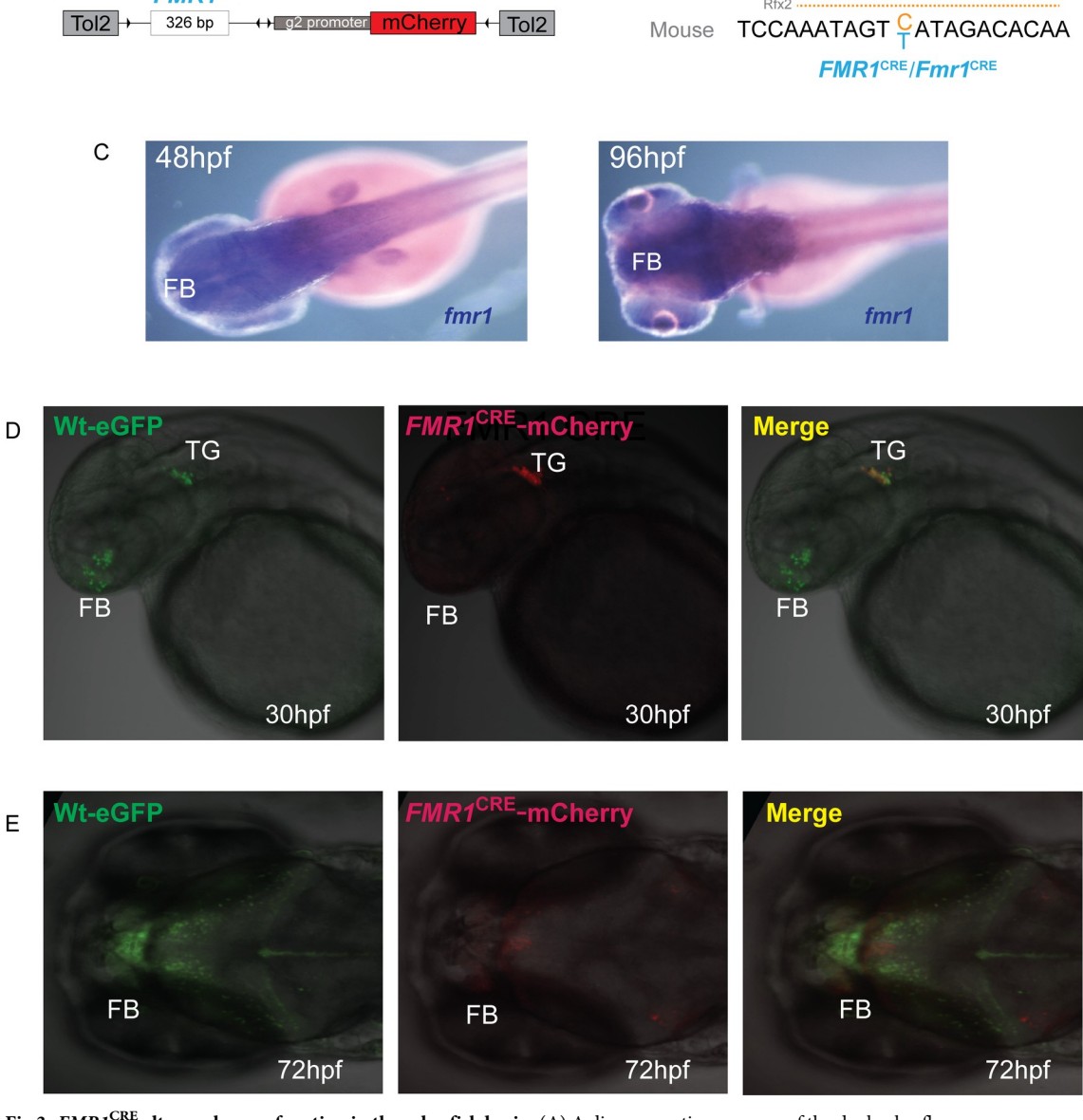

**Fig 3. *FMR1*CRE alters enhancer function in the zebrafish brain.** (A) A diagrammatic summary of the dual color fluorescence assay plasmid constructs used in this study. The size of the human *FMR1* element is provided in base pairs (bp) (B) Human and mouse (*FMR1*CRE/*Fmr1*CRE) sequences are shown with the variant base marked in blue, resulting in predicted loss of a RFX2/Rfx2 binding site in *FMR1*CRE/*Fmr1*CRE. (C) *mRNA in situ* hybridization showing expression of *fmr1* in forebrain and midbrain during embryonic development in wild-type zebrafish. (D-E) Dual color fluorescent transgenic assay in zebrafish with mutant *FMR1*CRE driving eGFP and mCherry expression respectively. Loss of enhancer activity is observed in forebrain with the mutant *FMR1*CRE allele. Further examples of embryos for different stable lines are shown in **S35 Fig in S1 File**. FB: Forebrain; MB: Midbrain; TG: Trigeminal ganglia; NP: hpf: Hours post fertilization.

injections. The amount of morpholino was titrated to the point where there was no morphological anomaly seen at 24 hours. When this concentration of morpholino was injected into *TENM1*WT/*TENM1*CRE transgenic embryos there was rescue of the activity of mutant CRE in the midbrain and hindbrain with no effect on the wild-type reporter (**Fig 2F and 2G**) and

knockdown of Six3 protein was confirmed by western blotting (**S40 Fig and S6 Note in S1 File**). This supports the hypothesis that the CRE variant had created a repressive SIX3 binding site as the mechanism for the transcriptional effect in zebrafish embryos.

The loss of a RFX2 binding site in both human and mouse *FMR1/Fmr1* CRE was predicted with relatively low confidence (**Fig 3B**). RFX2 (Regulatory factor X2) is a transcription factor required for spermatogenesis in mice and a wider role in the control of ciliogenesis [30–32]. Given the low confidence of this prediction we did not attempt any functional validation.

## *Fmr1*<sup>CRE</sup> and *Tenm1*<sup>CRE</sup> mouse models

CRISPR/Cas9 induced homologous recombination in mouse zygotes allowed us to individually "knock-in" the same nucleotide change identified in human *FMR1*<sup>CRE</sup> and *TENM1*<sup>CRE</sup> into the orthologous positions in the mouse genome (**Fig 1A**). We established multiple independent mouse lines for each CRE variant on a C57BL/6 background. All lines resulted in viable hemizygous mutant animals, at the expected ratio that were healthy and fertile with no obvious morphological abnormalities.

Whole-mount *in situ* hybridization (WISH) with riboprobes targeting either *Fmr1* or *Tenm1* was used to compare developmental expression patterns between wild-type and mutant male 13.5 gestational day (GD) embryos. *Fmr1*<sup>CRE</sup> caused a significant reduction in *Fmr1* expression in the olfactory placodes and the forebrain (**Fig 4A and 4B**). *Fmr1* WISH on four other wild-type and *Fmr1*<sup>CRE</sup> embryos is shown in **S41 Fig in S1 File**. *Tenm1*<sup>CRE</sup> did not show a consistent effect on *Tenm1* expression in male embryos at 13.5GD.

To determine if there were measurable phenotypic effects segregating with either CRE variant we first tested olfaction. This sense was selected for two reasons. First, complete loss of *Fmr1* expression in the olfactory placode in *Fmr1*<sup>CRE</sup> embryos was observed. Secondly, mutations in *TENM1/Tenm1* have recently been identified in humans and mice associated with congenital generalized anosmia [33]. Using a buried chocolate button test *Fmr1*<sup>CRE</sup> mice showed a significant increase in time to discovery compared to wild-type male littermates (**Fig 5A**). *Tenm1*<sup>CRE</sup> mice had olfactory function similar to wild-type male littermates (**Fig 5B**).

## *Fmr1*/FMRP-focused analyses

Loss of *FMR1* expression is responsible for Fragile X syndrome, the most common form of XLID [34]. Although we detected clear differences in *Fmr1* expression in embryonic midbrain and nasal placodes (**Fig 4A and 4B** and **S41 Fig in S1 File**), we did not find significant difference in *Fmr1* levels in the post-natal brains of male animals by quantitative RTPCR at postnatal day 7 (P7) or P14 (**S42 Fig in S1 File**). Similarly, at P25 we found no difference in *Fmr1* levels in forebrain, midbrain or hindbrain using mRNA sequencing (**S43 Fig in S1 File**) or in the ratio of *Fmr1:Pax6* transcripts in different regions of the hippocampus using *in situ* hybridization with dual-color RNAScope probe sets (**Fig 4E and 4F**).

Given the gene expression results, we were surprised to find an apparent increase in FMRP (fragile X mental retardation protein*)* protein abundance in the hippocampus of *Fmr1*<sup>CRE</sup> male mice compared to wild-type littermates using western blotting (**Fig 4G and 4H**). We found a decrease in mGluR-dependent long-term depression (LTD) in the CA3-CA1 hippocampus of *Fmr1*<sup>CRE</sup> males (**Fig 6A and 6B**). We considered the decrease in LTD to be consistent with the increased levels of FMRP protein given that an exaggerated LTD is a consistent finding in *Fmr1*-null animals [35]. A predisposition to audiogenic seizures is also a consistent phenotype in *Fmr1*-null mice but *Fmr1*<sup>CRE</sup> male mice showed no increase in such seizures (**Fig 6C**). The finding of a significant increase in bulk protein translation levels in the hippocampus of *Fmr1*<sup>CRE</sup> male mice (**Fig 6D**) was unexpected as this too is considered to be a marker of loss of *Fmr1* function [27].

Re-evaluation of affected individuals within the family in which *FMR1*^CRE^ is segregating (**Fig 7**) revealed no clinical features suggestive of a Fragile X (FRAX) syndrome diagnosis (OMIM #300624]; *FMR1* silencing) other than macrocephaly and intellectual disability. Importantly none of the individuals carrying *FMR1*^CRE^ showed signs of FRAX Tremor and Ataxia Syndrome (FRAXTAS [OMIM #300623]; *FMR1* over-expression) [36]. There was no obvious olfaction anomalies in the affected individuals from this family and no seizure predisposition. Clinical whole genome sequencing [17] of individual S3 (*FMR1*^CRE^ proband) did not identify any other plausible cause of his intellectual disability.

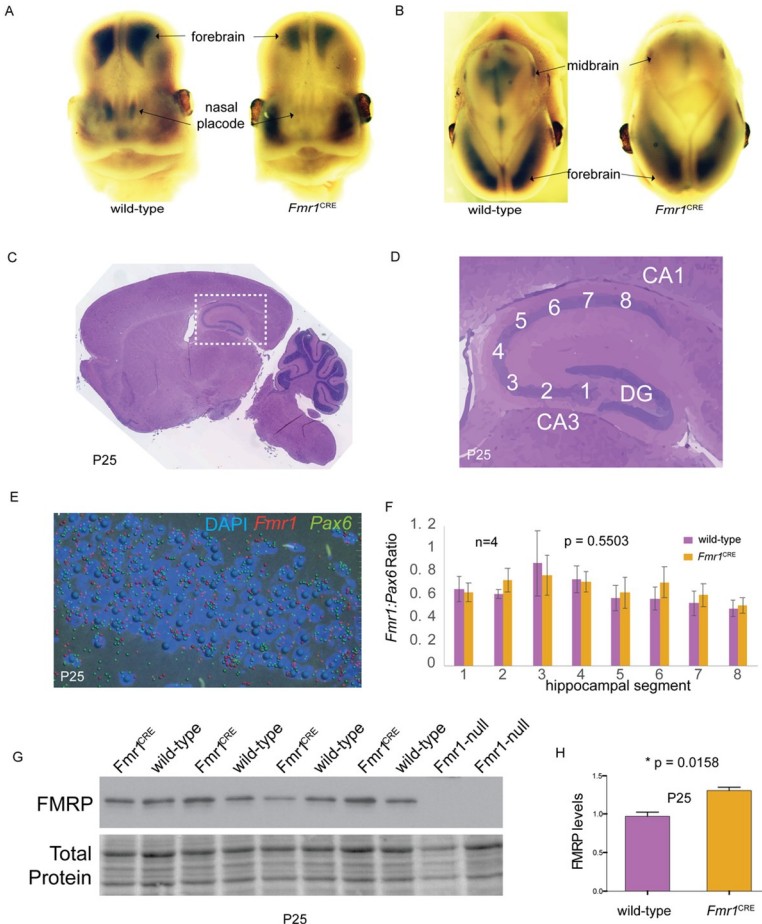

**Fig 4. Expression levels of *Fmr1* and FMRP in *Fmr1*^CRE^.** Frontal (A) and saggital (B) views of 13.5GD embryonic mouse heads following whole-mount *in situ* hybridization for *Fmr1*. In each panel the wild-type male embryo is shown on the left and the *Fmr1*^CRE^ embryo on the right. There is loss of expression of *Fmr1* in the nasal placode and midbrain *Fmr1*^CRE^ mutant embryos as compared to wild-type embryos. The *Fmr1*^CRE^ embryos had been deliberately over-developed in the chromogenic substrate compared to the wild-type embryos to emphasize the signal difference. Saggital H&E stained section of whole brain (C) with detailed view (white dashed box) of the hippocampus (D) with marked hippocampus regions indicating the regions analysed in (F) numbered 1–8, starting from dentate gyrus. (E) Reference image of RNAscope processed section with *Fmr1* transcript (red), *Pax6* transcript (green) and nucleus (blue/DAPI). Each transcript is represented by a spot following the quantitative image processing. (F) Graphical representation of *Fmr1* transcripts normalised to *Pax6* transcripts (used as control) between *Fmr1*^CRE^ (purple) compared to wild-type littermates (orange) and data represent average of four replicates (n = 4) ±SE. Levels of significance were determined by 2-tailed Student's t-test, with p values lower than 0.05 considered statistically significant. No significant difference was observed in the *Fmr1* transcript levels. (G) Western blot of hippocampal tissue from four *Fmr1*^CRE^, four wild-type and two *Fmr1*-null mice at P25 using an antibody that detects FMRP. (H) Quantitation of the FMRP bands in (G) indicating an apparent increase in FMRP in *Fmr1*^CRE^ hippocampal slices. All quantitative data are presented as mean ±SE and p value of 0.05 or less is considered statistically significant. (* means difference is statistically significant). FB: Forebrain; MB: Midbrain; NP: Nasal placode; DG: Dentate gyrus.

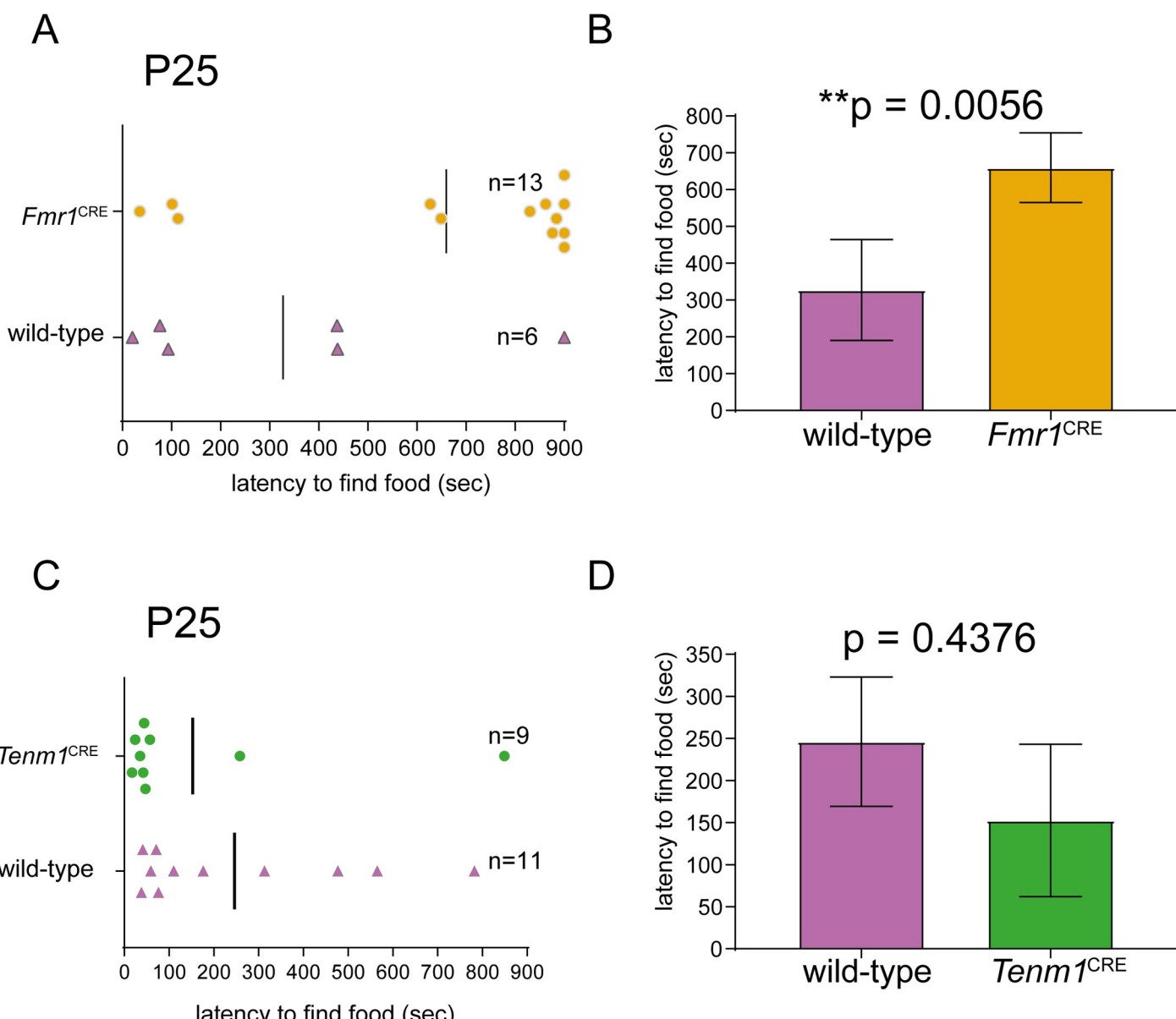

**Fig 5. Olfaction testing of *Fmr1*<sup>CRE</sup> and *Tenm1*<sup>CRE</sup> mouse lines.** (A, B) The mice hemizygous for the variant in *Fmr1*<sup>CRE</sup> showed a significant increase in time to discovery compared to wild-type male controls in a buried food test. (C, D) No significant difference in the levels of latency to find food was observed in mice hemizygous for the variant in *Tenm1*<sup>CRE</sup> compared to wild-type littermates. The numbers of animals tested (n) are given in (A) and (C). All quantitative data are presented as mean ±SE and p value of 0.05 or less is considered statistically significant. (* means difference is statistically significant).

Taken together the data above strongly suggest that *Fmr1*<sup>CRE</sup>/*FMR1*<sup>CRE</sup> does not result in simple loss or gain of *FMR1* function but rather a complex site and stage specific misregulation of gene product levels and cellular function.

## The impact of gnomAD 2.1 on the interpretation of CRE variants

The release of gnomAD 2.1 in late 2018 [37] represented a very significant change in our knowledge of the population allele frequencies in the non-coding part of the human genome. By this point we had already performed our zebrafish dual-colour transgenic screen and

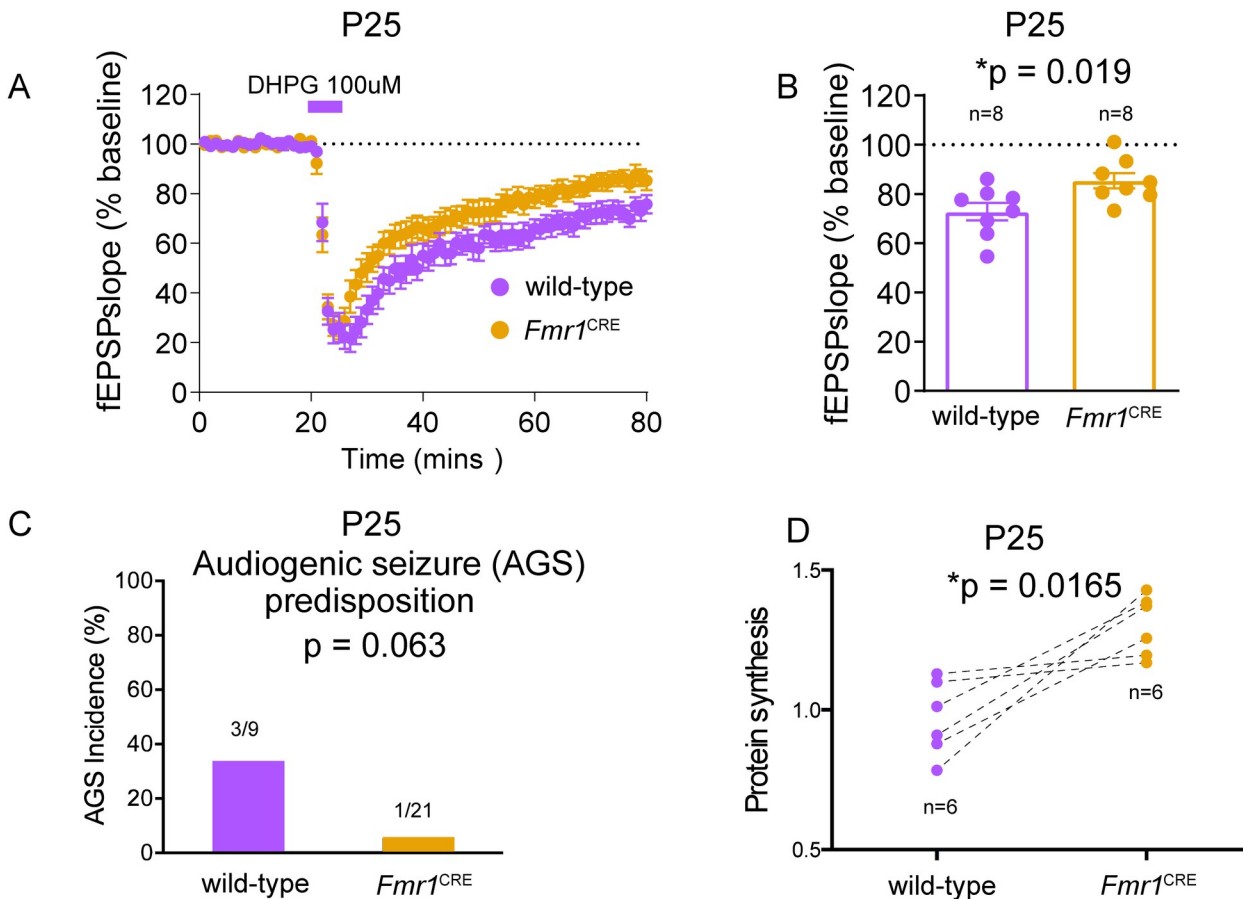

**Fig 6. Functional analysis of *FMR1*^CRE^ mice.** (A) Comparison of mGluR-dependent long-term depression (LTD) in CA3-CA1 components of the hippocampus of eight *Fmr1*^CRE^ male mice and eight wild-type male littermates indicates a significant *Fmr1*^CRE^-associated decrease in LTD. (B) All quantitative data are presented as mean ±SE and p value of 0.05 or less is considered statistically significant.(C) No significant difference was observed in audiogenic seizure incidence in the hemizygous mice with the variant *Fmr1*^CRE^(1/21) compared to wild-type littermates(3/9). Statistical significance is determined using two-tailed Fisher's exact test and p value of 0.05 or less is considered statistically significant.(D) Significant increase in bulk protein synthesis levels in slices from dorsal hippocampus of *Fmr1*^CRE^ knock-in mutant male mice as compared to wild-type male littermates. Quantitative data is derived from number of biological replicates used (n = 6) in the experiments. Levels of significance were determined by 2-tailed Student's t-test, with p values lower than 0.05 considered statistically significant. (* means difference is statistically significant).

created the mouse models for *Tenm1*^CRE^ and *Fmr1*^CRE^. The gnomAD-derived variant allele frequencies (AF) of the six variants which survived our original filtering are shown in **Table 1**. This showed three of the variants remained unique; *FMR1*^CRE^, *POLA1/PCYT1B*^CRE^ (DNA polymerase alpha 1, catalytic subunit) and *KDM6A*^CRE^ (Lysine demethylase 6A). However, *TENM1*^CRE^, *ARHGEF6*^CRE^ (Rac/Cdc42 guanine nucleotide exchange factor 6) and *AFF2*^CRE^ (AF4/FMR2 family member 2) were observed in the gnomAD population and the latter two variants were also seen in hemizygous state suggesting that they were very unlikely to be a cause of XLID. Although the allele frequency of *TENM1*^CRE^ was below 1 in 10,000, a frequency commonly used for clinical filtering of ultrarare variants, we wished to know if this should change its "plausibly causative" status. Using the approach of Whiffin et al [38] we calculated the maximum plausible allele frequency (AF) for any XLID causal variant. We chose conservative parameters: 0.01 for genetic heterogeneity (i.e. 1% of all XLID without a known diagnosis is caused by variation in a particular CRE), 0.2 for allelic heterogeneity (i.e. only 5 different causative variants can exist per CRE) and 0.5 for penetrance (this is complicated by X-linked

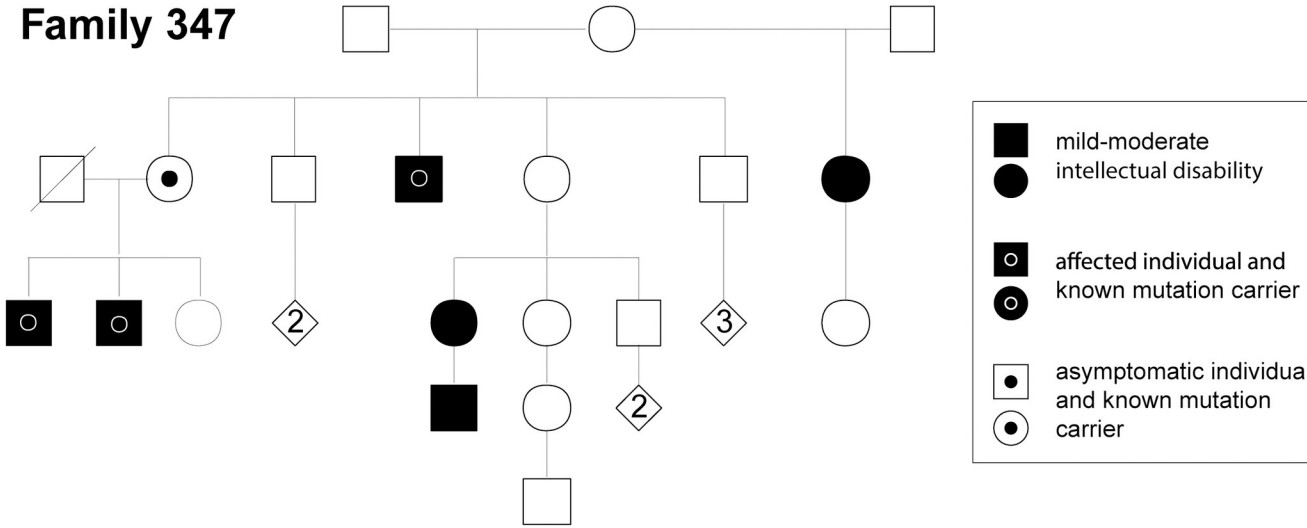

**Fig 7. Family 347 pedigree.** Pedigree of Family 347 of which individual S3 is a member showing segregation of the mutation affecting *FMR1* expression.

inheritance but likely to be ~1 in males and $>$ = 0.1 in females). These parameters gave maximum permitted 95% confidence AF = 4e06 suggesting that *TENM1*^CRE is not a plausibly cause of XLID.

## Discussion

The motivation for initiating this study was the difficulty in assigning pathogenic or likely pathogenic status to any *de novo* or segregating variant in an intergenic regulatory sequence. Such ultra-rare variants would be almost universally be considered of uncertain significance using current best practice guidelines for diagnostic interpretation of genomic sequence variants [39,40]. However functional assays demonstrating that an abnormality gene function associated with a CRE variant (coded as PS3 in the ACMG guidelines) has the potential to change many variants of uncertain significance (VUS) to likely pathogenic status [41]. The question then becomes: how should we use data from functional assays in clinical interpretation of regulatory variants. Given the rapid switch from targeted whole exome sequencing to whole genome sequencing, it is likely that there will be an increasing need to develop a rational approach to the interpretation of ultra-rare regulatory variants.

Here we performed an integrated clinical, genetic, developmental, behavioural and neurophysiological approach to the analyses of CRE variants identified in a cohort of affected individuals that should be enriched for causative cis-regulatory mutations. XLID accounts for ~16% of ID in males [42]. Mutations in the coding region of at least 81 different genes [16,43] have been identified as causing XLID. Given the significant contribution of XLID to ID and the observed regulatory variant enrichment in a large cohort of individuals with neurodevelopmental disorders [13] we reasoned that we could increase the prior probability of identifying likely causative mutations by restricting the genomic search space to the X chromosome. We also chose to limit our investigations to variants in enhancers targeting known XLID genes, since most of the known disease-associated regulatory mutations were identified because they partially [44] or fully [45] phenocopy loss-of-function mutations in the target gene. If this were true for our cohort, then matching the pattern of clinical features of individuals carrying a specific regulatory mutation to those of the syndrome associated with intragenic mutations would have diagnostic value.

Our experimental design can be justified on pragmatic grounds. However, we do recognize some significant problems with this filtering strategy. First, the CNE variant could induce expression in cells types where the target gene is normally silenced, which is likely to result in a phenotype completely distinct from that associated with intragenic mutations. Secondly, if intragenic mutations in the target gene result in early embryonic lethality this gene would not be identified as a cause of XLID. However, mutations in a CRE controlling only neural expression of an essential gene could cause XLID but would be missed in our analysis which is focussed on known XLID genes. Recently somatic mutations in the brain have been implicated in the causation of neuro developmental disorders [46]. The fact that we have selected individuals with a positive family history would significantly reduce the prior probability of this disease mechanism in our cohort. For this reason, we have designed our variant filtering strategy to identify constitutional mutations and exclude variants that are likely to be mosaic.

*In vivo* analysis of the enhancer activity using the dual color reporter transgenic zebrafish embryos proved to be a useful screen. Four CRE (*POLA1-PCYT1B*, *ARHGEF6*, *KDM6A*, *AFF2*) showed inconsistent and/or non-specific reporter activity with no difference between the wild-type and mutant alleles (**Table 2**). However, this analysis also provided evidence for abnormal in developmental gene regulation for two CRE controlling *FMR1* and *TENM1* respectively. For *TENM1*<sup>CRE</sup> we could identify the mechanism of the altered reporter gene expression in zebrafish: *de novo* formation of a repressive six3 binding site.

It is remarkable that, in absence of obvious homology with human CRE sequences (**S44 Fig in S1 File**), the developing zebrafish brain appears to report the same transcription factors to control site and stage specific gene activation. This argues for a subtler shape-based interaction of DNA with transcription factors that we are, as yet, unable to understand. Defining the grammar of this structural effect will significantly aid our interpretation of variants in the non-coding genome.

The unique CRE variant *FMR1*<sup>CRE</sup> is the most plausible disease associated allele of those identified in this study. This variant produced abnormal embryonic expression of endogenous *Fmr1* in a mouse model (**Fig 4A and 4B, S40 Fig in S1 File**) consistent with the tissue specific loss of function during early brain development in the transgenic zebrafish embryos. In contrast, we were unable to show evidence of altered transcriptional regulation in the post-natal brain of *Fmr1*<sup>CRE</sup> male mice (**Fig 4C–4F**). This was particularly interesting given the apparent increased level of FMRP protein in hippocampal slices derived from P25 *Fmr1*<sup>CRE</sup> mice (**Fig 4G and 4H).** This increase may explain the electrophysiological change of LTD we observed being the opposite effect to that seen in *Fmr1* KO mice. The increase in bulk protein synthesis was surprising as this effect is seen in *Fmr1* KO mice. These apparently paradoxical results are likely to reflect a complex developmental mis-programming of the cells in the hippocampus.

The results outlined above, provide a clear explanation for why proband S3 and his affected male relatives carrying *FMR1*<sup>CRE</sup>, do not show a clinical pattern typical of either Fragile X syndrome [OMIM 300624] or FRAXTAS [OMIM 300623]. The family presented with a non-specific intellectual disability associated with mild macrocephaly. We consider it likely that many causative CRE variants be associated with clinical features that differ significantly from those seen associated with intragenic mutations of target gene. This means that we have relatively limited ability to predict the phenotypes associated with regulatory mutations even when the clinical impact of intragenic mutations of target gene are well characterised.

While it remains challenging to recognise causative CRE variants, the availability of population-based allele frequencies from whole genome sequencing data has certainly improved our ability to identify those of likely neutral or low penetrant effect. The gnomAD 2.1 data allowed us to show that *TENM1*<sup>CRE</sup> was implausible as an XLID causative variant despite it being in an evolutionarily conserved, non-redundant CRE with a strong repressive effect in the zebrafish

transgenic analyses. Filtering for extreme rarity of individual alleles will aid the identification of causative variants in CRE that are under high levels of selective constraint within human populations [47]. However, human genetic filtering will have to be supported by strong, multi-source, disease-relevant functional data before a likely causative status can be assigned to any CRE variant.

## Supporting information

**S1 Table. List of coordinates used to design customized capture library.**
(TXT)

**S1 File. The file have details for S1-S44 Figs, S1-S3 Tables and S1-S6 Notes.**
(DOCX)

**S1 Raw images.**
(PDF)

## Author Contributions

**Conceptualization:** Veronica van Heyningen, F. Lucy Raymond, Hugues Roest Crollius, David R. FitzPatrick.

**Data curation:** Hemant Bengani, Lambert Moyon, James G. Prendergast, Graeme Grimes, F. Lucy Raymond, Hugues Roest Crollius.

**Formal analysis:** Hemant Bengani, Lambert Moyon, James G. Prendergast, Liusaidh J. Owen, Magali Naville, Jacqueline Rainger, Graeme Grimes, Mihail Halachev, Laura C. Murphy, Olivera Spasic-Boskovic, Hugues Roest Crollius.

**Funding acquisition:** Hemant Bengani, F. Lucy Raymond, Hugues Roest Crollius, David R. FitzPatrick.

**Investigation:** Hemant Bengani, Shipra Bhatia.

**Methodology:** Hemant Bengani, Shipra Bhatia, Susana R. Louros, Jilly Hope, Adam Jackson, David R. FitzPatrick.

**Resources:** Detelina Grozeva, Olivera Spasic-Boskovic, Peter Kind, Catherine M. Abbott, Emily Osterweil, F. Lucy Raymond.

**Software:** Lambert Moyon.

**Supervision:** Hugues Roest Crollius, David R. FitzPatrick.

**Validation:** Hemant Bengani, Detelina Grozeva, Shipra Bhatia, Susana R. Louros, Jilly Hope, Adam Jackson, Jacqueline Rainger.

**Visualization:** Hemant Bengani, Shipra Bhatia, Susana R. Louros, Jilly Hope, Adam Jackson, Jacqueline Rainger, Graeme Grimes, David R. FitzPatrick.

**Writing – original draft:** Hemant Bengani, Shipra Bhatia, David R. FitzPatrick.

**Writing – review & editing:** Hemant Bengani, Veronica van Heyningen, Peter Kind, Catherine M. Abbott, Emily Osterweil, F. Lucy Raymond, Hugues Roest Crollius, David R. FitzPatrick.

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
