## [Decision Letter · Decision Letter 0]

12 May 2021

PONE-D-21-12396

Functional predictors of causation for cis-regulatory mutations

PLOS ONE

Dear Dr. FitzPatrick,

Thank you for submitting your manuscript to PLOS ONE. After careful consideration, we feel that it has merit but does not fully meet PLOS ONE’s publication criteria as it currently stands. Therefore, we invite you to submit a revised version of the manuscript that addresses the points raised during the review process.

We look forward to receiving your revised manuscript.

Kind regards,

Chaeyoung Lee

Academic Editor

PLOS ONE

Journal Requirements:

2. To comply with PLOS ONE submissions requirements, in your Methods section, please provide additional information on the animal research and ensure you have included details on (1) methods of sacrifice, (2) methods of anesthesia and/or analgesia, and (3) efforts to alleviate suffering.

3. We note that you are reporting an analysis of a microarray, next-generation sequencing, or deep sequencing data set. PLOS requires that authors comply with field-specific standards for preparation, recording, and deposition of data in repositories appropriate to their field. Please upload these data to a stable, public repository (such as ArrayExpress, Gene Expression Omnibus (GEO), DNA Data Bank of Japan (DDBJ), NCBI GenBank, NCBI Sequence Read Archive, or EMBL Nucleotide Sequence Database (ENA)). In your revised cover letter, please provide the relevant accession numbers that may be used to access these data. For a full list of recommended repositories, see http://journals.plos.org/plosone/s/data-availability#loc-omics or http://journals.plos.org/plosone/s/data-availability#loc-sequencing.

Reviewers' comments:

Reviewer's Responses to Questions

**Comments to the Author**

1. Is the manuscript technically sound, and do the data support the conclusions?

Reviewer #1: Yes

Reviewer #2: Yes

Reviewer #3: Yes

2. Has the statistical analysis been performed appropriately and rigorously? 

Reviewer #1: Yes

Reviewer #2: N/A

Reviewer #3: Yes

3. Have the authors made all data underlying the findings in their manuscript fully available?

Reviewer #1: Yes

Reviewer #2: Yes

Reviewer #3: Yes

4. Is the manuscript presented in an intelligible fashion and written in standard English?

Reviewer #1: Yes

Reviewer #2: Yes

Reviewer #3: Yes

5. Review Comments to the Author

Reviewer #1: This manuscript by Bengani and FitzPatrick used an updated method to determine causality of SNPs found in genomes of patients with genetically inherited intellectual disability (ID). Those are resourceful information to the other peers studying X-linked recessive forms of ID and represent diagnostic values for this particular disorder. The data in this work use a combination of zebra fish and mouse genetics models and functionally validated the clinically identified rare variants on the X chromosomes. Overall, I agree that the study is conclusive and thus recommend for acceptance.

I list two minor issues and I hope the authors can address them quickly before publication.

1, An increasing body of evidences have implied brains are highly associated with somatic mutations, which often cause defects either derived from neural degeneration or development. Dr. Christopher Walsh from Harvard University is an expert in this field. The point is brain genomes are not identical to genomes of other tissue types as the method presented in this study used nonbrain tissues for sequencing. Would the authors comments on this point with respect to performance of gnomAD2.1 in the discussion section?

2, The quality/resolution of some figures are low, e.g. fig. 1, 3, 4 and 5. Is there any way to increase the readability of these figures?

Reviewer #2: Bengani et al. performed a study for the identification of functional cis-regulatory mutations involved in the development of X-linked intellectual disability. The task is definitely very interesting, but also very challenging and time consuming. Indeed, it took several years and two different in vivo models to find data supporting a causative role for one out of 31 highly conserved CRE initially identified, the FMR1-CRE variant.

I think that this manuscript is worth publishing because it has a quite strong rationale behind the selection of the non-coding variants to be tested and it shows that it is possible to make functional sense of mutations in regulatory elements. However, the phenotype for which the authors are trying to find a genetic evidence in CRE is very complex and very hard to dissect. Indeed, it is very difficult to find a straightforward link with a single variant in CRE. The phenotypes observed in the mouse model and the link to the human disease appears quite weak. I think this should be clearly underlined in the manuscript and better explained.

Moreover, I think that the title, being very general, overstates the results of the manuscript and it should be changed to make it more related to the findings.

The description of the statistical analysis applied by the authors is lacking. Moreover, I would suggest to check the statistical methods used for the analysis, in general, and, in particular, for Fmr1 transcription by qPCR (Fig. S41): some differences between wt and mutants would seem significant (i.e. in Hind brain P7), especially compared to other data reported as significant in the manuscript (i.e. Fig. 4H). Based on these reevaluations, some results and some statement in the paper might need to be revised.

Reviewer #3: Manuscript Summary:

Identifying causative variants in cis -regulatory elements (CREs) in neurodevelopmental

disorders has proven challenging. Bengani, Grozeva, Moyon, and Bhatia et al. used a translational bridge between humans-zebrafish-mice, with the aid of model systems, and circling back to humans to identify and evaluate the pathological implications of chromosome X-resident CRE mutations that cause X-linked intellectual disability (XLID).

Overall, this manuscript is perfectly suited for the scope of PLOS ONE. The translational approach that the authors have taken in this study is albeit challenging but particularly well appreciated. Studies such as these would certainly enhance the field's understanding of pathogenic variants implicated in neurodevelopmental disorders. In addition, the methodology employed in this study is thorough and the authors present the data/findings objectively.

Major Comments and Questions:

1)

1a) Figures 2F and 2G: The authors use Six3-targeting morpholinos to demonstrate that six3-knockdown rescues the activity of mutant TENM1-CRE, the latter of which is quite convincing. However, there is no validation data provided to demonstrate that the concentration of the six3-morpholino used post-optimizations indeed led to a sufficient knockdown of six3.

1b) Additional note 1 (not required for the acceptance of this manuscript for publication): While I can understand the following might be outside the scope of this manuscript, an IP-seq experiment to pull down six3-bound to either TENM1-WT or TENM1-CRE would provide irrefutable evidence to substantiate the author's hypothesis that TENM1-CRE introduces a novel binding site of the repressor, six3.

1c) Additional note 2: The following could be addressed in the discussion, as per the authors' discretion. While TENM1-CRE led to a consistent loss of tenm1-expression in the mid-brain and hind-brain (table 1), it did not lead to a loss of tenm1-expression in the neural tube. Is six3 and six6 not expressed in the neural tubes and what additional six3/six6-independent mechanisms could be at play here?

2) In the figure legend for Fig. 6C: The authors state that there was "No significant difference was observed

in audiogenic seizure incidence in the hemizygous mice with the variant Fmr1CRE compared to wild-type littermates (lines 447 - 449)." However, as per Fig. 6C, the Fmr1-CRE did display a statistically-significant decrease in the predisposition to audiogenic seizures (AGS), starkly contrasting the anticipated results of an increase in AGS derived from Fmr1-null mice. The data and results appears to contradict the figure legend and the authors' interpretation.

3) It is nice that the authors took a bench-to-bedside approach to re-evaluate the individuals within the FMR1-CRE-segregating family. Did these individuals not have any olfactory disorders, which may recapitulate the phenotype observed in Fig. 5A,B? And, following up from the precedence derived from Fig. 6C, did the affected individuals also display a resistance to the development of AGS?

4) The six3 piece of data from the zebrafish study was intriguing (Fig. 2). Was there no correlative evidence of a similar interplay between Six3 and Tenm1 in mice and/or humans? Could murine and human SIX3 exhibit a non-overlapping pattern of expression with TENM1, which might explain the lack of a phenotype in Tenm1-CRE mice? What are the authors thoughts on the Six3/Six6-dependent and -independent mechanisms of tenm1-CRE in zebrafish and how these mechanisms translate to mammalian species (e.g. mice and humans)?

Minor Comments and Questions:

1) Formatting: It would be nice to keep the text format used in all figures, either in the main text or in the supplement, consistent in size and style.

2) It would be nice to spell out the full names of the protein abbreviations used, such as FMR1, FMRP, TENM1.

All additional major and minor comments from me were either fully or partially overlapping with those raised by the reviewers during the peer-review for the PLOS Genetics submission and have been addressed by the authors.

6. PLOS authors have the option to publish the peer review history of their article (what does this mean?). If published, this will include your full peer review and any attached files.

Reviewer #1: No

Reviewer #2: No

Reviewer #3: No

---

## [Author Response · Author response to Decision Letter 0]

21 Jul 2021

Reviewer #1: 

This manuscript by Bengani and FitzPatrick used an updated method to determine causality of SNPs found in genomes of patients with genetically inherited intellectual disability (ID). Those are resourceful information to the other peers studying X-linked recessive forms of ID and represent diagnostic values for this particular disorder. The data in this work use a combination of zebra fish and mouse genetics models and functionally validated the clinically identified rare variants on the X chromosomes. Overall, I agree that the study is conclusive and thus recommend for acceptance.

I list two minor issues and I hope the authors can address them quickly before publication.

1, An increasing body of evidences have implied brains are highly associated with somatic mutations, which often cause defects either derived from neural degeneration or development. Dr. Christopher Walsh from Harvard University is an expert in this field. The point is brain genomes are not identical to genomes of other tissue types as the method presented in this study used nonbrain tissues for sequencing. Would the authors comments on this point with respect to performance of gnomAD2.1 in the discussion section?

This is an interesting point. Our experimental design relies on the variants being inherited and co-segregating with the disease in an individual family. gnomAD, likewise, is very focused on cataloging constitutional variation and data is filtered with the specific purpose of excluding somatic mosaic variants. To clarify this we have added the following text to the discussion:

 “Recently somatic mutations in the brain have been implicated in the causation of neuro developmental disorders {Rodin et al., 2021, #61844}. The fact that we have selected individuals with a positive family history would significantly reduce the prior probability of this disease mechanism in our cohort. For this reason, we have designed our variant filtering strategy to identify constitutional mutations and exclude variants that are likely to be mosaic.” (Line: 460-464)

2, The quality/resolution of some figures are low, e.g. fig. 1, 3, 4 and 5. Is there any way to increase the readability of these figures?

We apologise for this and we have increased the resolution of these figures – we also have Adobe Illustrator files available if required.

 

Reviewer #2: 

Bengani et al. performed a study for the identification of functional cis-regulatory mutations involved in the development of X-linked intellectual disability. The task is definitely very interesting, but also very challenging and time consuming. Indeed, it took several years and two different in vivo models to find data supporting a causative role for one out of 31 highly conserved CRE initially identified, the FMR1-CRE variant.

I think that this manuscript is worth publishing because it has a quite strong rationale behind the selection of the non-coding variants to be tested and it shows that it is possible to make functional sense of mutations in regulatory elements. However, the phenotype for which the authors are trying to find a genetic evidence in CRE is very complex and very hard to dissect. Indeed, it is very difficult to find a straightforward link with a single variant in CRE. The phenotypes observed in the mouse model and the link to the human disease appears quite weak. I think this should be clearly underlined in the manuscript and better explained.

We have tried not to over-state the link between the mouse model and the human disease. However for the FMR1CRE/Fmr1CRE variant there is clear evidence that this variant results in reduced expression of endogenous Fmr1 in the embryonic brain and abnormal results in neurophysiological tests that are considered to the gold-standard markers of FMRP function in the hippocampus. We agree that the behavioural phenotype is not Fmr1-specific but there are no known phenotypes that are specific to this genotype. We have changed the penultimate sentence in the abstract to address the reviewer’s concerns. It now reads:

”Assigning causative status to any ultra-rare CRE variant remains problematic and requires disease-relevant in vivo functional data from multiple sources.”(Line: 53-55).

Moreover, I think that the title, being very general, overstates the results of the manuscript and it should be changed to make it more related to the findings.

We have changed the title to be more descriptive: 

“Identification and functional modelling of plausibly causative cis-regulatory variants in a highly-selected cohort with X-linked intellectual disability.”

The description of the statistical analysis applied by the authors is lacking. 

Moreover, I would suggest to check the statistical methods used for the analysis, in general, and, in particular, for Fmr1 transcription by qPCR (Fig. S41): some differences between wt and mutants would seem significant (i.e. in Hind brain P7), especially compared to other data reported as significant in the manuscript (i.e. Fig. 4H). Based on these reevaluations, some results and some statement in the paper might need to be revised.

We have added a section named Statistical analysis (Line 255-260) under Methods in which we have given details of statistical methods used and also added the details of statistics used in the legend section of figures. There was mistake in calculating the standard error in figure S41.We have re done the analysis and updated the figure S41 (now S42 Fig) and result confirm that differences between wt and mutants is not significant. We apologise for this error.

 

Reviewer #3: Manuscript Summary:

Identifying causative variants in cis -regulatory elements (CREs) in neurodevelopmental

disorders has proven challenging. Bengani, Grozeva, Moyon, and Bhatia et al. used a translational bridge between humans-zebrafish-mice, with the aid of model systems, and circling back to humans to identify and evaluate the pathological implications of chromosome X-resident CRE mutations that cause X-linked intellectual disability (XLID).

Overall, this manuscript is perfectly suited for the scope of PLOS ONE. The translational approach that the authors have taken in this study is albeit challenging but particularly well appreciated. Studies such as these would certainly enhance the field's understanding of pathogenic variants implicated in neurodevelopmental disorders. In addition, the methodology employed in this study is thorough and the authors present the data/findings objectively.

Major Comments and Questions:

1)

1a) Figures 2F and 2G: The authors use Six3-targeting morpholinos to demonstrate that six3-knockdown rescues the activity of mutant TENM1-CRE, the latter of which is quite convincing. However, there is no validation data provided to demonstrate that the concentration of the six3-morpholino used post-optimizations indeed led to a sufficient knockdown of six3.

We have validated the knockdown of Six3 by western blot. We have added a figure in the supplementary data named as S40 Fig and also documented it in the result section.(Line 333-334)

1b) Additional note 1 (not required for the acceptance of this manuscript for publication): While I can understand the following might be outside the scope of this manuscript, an IP-seq experiment to pull down six3-bound to either TENM1-WT or TENM1-CRE would provide irrefutable evidence to substantiate the author's hypothesis that TENM1-CRE introduces a novel binding site of the repressor, six3.

We thank the reviewer for this suggestion, and we would endeavour to do those experiments in future follow-up studies. 

1c) Additional note 2: The following could be addressed in the discussion, as per the authors' discretion. While TENM1-CRE led to a consistent loss of tenm1-expression in the mid-brain and hind-brain (table 1), it did not lead to a loss of tenm1-expression in the neural tube. Is six3 and six6 not expressed in the neural tubes and what additional six3/six6-independent mechanisms could be at play here?

We believe there might be distinct regions of the TENM1-CRE driving expression in the mid-brain/hind-brain and the neural tube. The single-nucleotide change affecting the six3 binding site possibly localises to the region of the CRE driving brain expression. Similar scenarios have been observed in other CREs (Bhatia et al AJHG 2013)

2) In the figure legend for Fig. 6C: The authors state that there was "No significant difference was observed

in audiogenic seizure incidence in the hemizygous mice with the variant Fmr1CRE compared to wild-type littermates (lines 447 - 449)." However, as per Fig. 6C, the Fmr1-CRE did display a statistically-significant decrease in the predisposition to audiogenic seizures (AGS), starkly contrasting the anticipated results of an increase in AGS derived from Fmr1-null mice. The data and results appears to contradict the figure legend and the authors' interpretation.

The apparent decrease in audiogenic seizures in the Frm1-CRE mice is intriguing but is not statistically significant. Significance was determined using two-tailed Fisher’s exact test (appropriate for analysing nominal data sets). A p-value of < 0.05 was considered statistically significant. The calculated p value is p = 0.063 which is greater than the cut off used. 

3) It is nice that the authors took a bench-to-bedside approach to re-evaluate the individuals within the FMR1-CRE-segregating family. Did these individuals not have any olfactory disorders, which may recapitulate the phenotype observed in Fig. 5A,B? And, following up from the precedence derived from Fig. 6C, did the affected individuals also display a resistance to the development of AGS?

There was no obvious olfaction anomalies in the affected individuals from this family and no seizure predisposition (Line 393-394). We have added this as a comment to the second last paragraph of the Results section “Fmr1/FMRP-Focused Analyses”.

4) The six3 piece of data from the zebrafish study was intriguing (Fig. 2). Was there no correlative evidence of a similar interplay between Six3 and Tenm1 in mice and/or humans? Could murine and human SIX3 exhibit a non-overlapping pattern of expression with TENM1, which might explain the lack of a phenotype in Tenm1-CRE mice? What are the authors thoughts on the Six3/Six6-dependent and -independent mechanisms of tenm1-CRE in zebrafish and how these mechanisms translate to mammalian species (e.g. mice and humans)?

We agree this it would be interesting to explore the relationship between SIX3-binding and TENM1 expression in the mouse mutants but this is obviously a much more challenging experimental task than it is in zebrafish embryos. The zebrafish reporter-transgenic assay used an isolated fragment of human DNA the dramatic effect of Six3 binding on CRE function may be exaggerated compared to the endogenous loci in mice and humans if other enhancers exist with partially overlapping function.

Minor Comments and Questions:

1) Formatting: It would be nice to keep the text format used in all figures, either in the main text or in the supplement, consistent in size and style.

We have reformatted all the text in the figures and legends in the main text as well as in supplement with consistent size and style. We were not able to change the format of figure S3-S33 (pedigree analysis) as we got the file from the clinical lab long time back and unable to access the original file. We apologise for this.

2) It would be nice to spell out the full names of the protein abbreviations used, such as FMR1, FMRP, TENM1.

We have added the details of full gene name/protein abbreviation in the main text 

FMR1: FMRP translational regulator 1. (Line 96)

Tenm1: Teneurin transmembrane protein 1. (Line 184)

SIX3: SIX homeobox 3. (Line 323)

 SIX6: SIX homeobox 6. (Line 324)

RFX2: Regulatory factor X2. (Line 338)

FMRP: fragile X mental retardation protein. (Line 375)

POLA1/PCYT1B: DNA polymerase alpha 1, catalytic subunit. (Line 408)

KDM6A: lysine demethylase 6A. (Line 409)

ARHGEF6: Rac/Cdc42 guanine nucleotide exchange factor 6. (Line 410)

AFF2: AF4/FMR2 family member 2. (Line 410)

All additional major and minor comments from me were either fully or partially overlapping with those raised by the reviewers during the peer-review for the PLOS Genetics submission and have been addressed by the authors.

---

## [Decision Letter · Decision Letter 1]

2 Aug 2021

Identification and functional modelling of plausibly causative cis-regulatory variants in a highly-selected cohort with X-linked intellectual disability

PONE-D-21-12396R1

Dear Dr. FitzPatrick,

We’re pleased to inform you that your manuscript has been judged scientifically suitable for publication and will be formally accepted for publication once it meets all outstanding technical requirements.

Kind regards,

Chaeyoung Lee

Academic Editor

PLOS ONE

Additional Editor Comments (optional):

Reviewers' comments:

Reviewer's Responses to Questions

**Comments to the Author**

1. If the authors have adequately addressed your comments raised in a previous round of review and you feel that this manuscript is now acceptable for publication, you may indicate that here to bypass the “Comments to the Author” section, enter your conflict of interest statement in the “Confidential to Editor” section, and submit your "Accept" recommendation.

Reviewer #1: All comments have been addressed

Reviewer #2: All comments have been addressed

Reviewer #3: All comments have been addressed

2. Is the manuscript technically sound, and do the data support the conclusions?

Reviewer #1: Yes

Reviewer #2: (No Response)

Reviewer #3: Yes

3. Has the statistical analysis been performed appropriately and rigorously? 

Reviewer #1: Yes

Reviewer #2: (No Response)

Reviewer #3: Yes

4. Have the authors made all data underlying the findings in their manuscript fully available?

Reviewer #1: Yes

Reviewer #2: (No Response)

Reviewer #3: Yes

5. Is the manuscript presented in an intelligible fashion and written in standard English?

Reviewer #1: Yes

Reviewer #2: (No Response)

Reviewer #3: Yes

6. Review Comments to the Author

Reviewer #1: All of my minor concerns have been addressed in this revisedmanuscript. I have no further question.

Reviewer #2: (No Response)

Reviewer #3: The authors have sufficiently and more than satisfactorily addressed all my comments and questions.

Overall, this manuscript is perfectly suited for the scope of PLOS ONE. The translational approach that the authors have taken in this study is albeit challenging but particularly well appreciated. Studies such as these would certainly enhance the field's understanding of pathogenic variants implicated in neurodevelopmental disorders. In addition, the methodology employed in this study is thorough and the authors present the data/findings objectively.

7. PLOS authors have the option to publish the peer review history of their article (what does this mean?). If published, this will include your full peer review and any attached files.

Reviewer #1: No

Reviewer #2: No

Reviewer #3: No

---

## [Editor Report · Acceptance letter]

5 Aug 2021

PONE-D-21-12396R1 

Identification and functional modelling of plausibly causative *cis*-regulatory variants in a highly-selected cohort with X-linked intellectual disability 

Dear Dr. FitzPatrick:

I'm pleased to inform you that your manuscript has been deemed suitable for publication in PLOS ONE. Congratulations! Your manuscript is now with our production department. 

Kind regards, 

on behalf of

Prof. Chaeyoung Lee 

Academic Editor

PLOS ONE